# Feasibility Study of Fluidic Sealing in Turbine Shroud

**DOI:** 10.3390/ma14133477

**Published:** 2021-06-22

**Authors:** Filip Wasilczuk, Pawel Flaszynski, Lukasz Pyclik, Krzysztof Marugi

**Affiliations:** 1Institute of Fluid-Flow Machinery, Polish Academy of Sciences, Fiszera 14, 80-952 Gdansk, Poland; pflaszyn@imp.gda.pl; 2Avio Polska Sp. z o.o., Grażyńskiego 141, 43-346 Bielsko-Biała, Poland; Lukasz.Pyclik@avioaero.it (L.P.); krzysztof.marugi@avioaero.it (K.M.)

**Keywords:** labyrinth seal, gas turbine, additive manufacturing, leakage reduction, flow control

## Abstract

This paper analyses the methods for manufacturing turbine blades, focusing on the possibility of manufacturing slots in the region of the shroud. The reason for this analysis is the new flow control technique that can be used to limit the shroud leakage flow in a turbine—the air curtain. The air curtain uses a bypass slot to connect the upstream cavern of a shroud seal with the tip of a shroud fin. The bypass slot is an essential part of the solution, while at the same time introducing difficulties in the manufacturing process. Additionally, a parametric study on the bypass slot dimensions is performed using numerical simulations. The features of the numerical model and its validation against experimental data are presented. The parametric study includes the inlet and outlet dimensions, as well as the width of the slot. The most effective dimensions are shown, along with a possible explanation as to why they are the most effective. Interestingly, a slot that does not cover the whole span of the fin is more effective than a slot covering the whole span of the fin. This is caused by additional streamwise vortices that are created in the proximity of the bypass slot.

## 1. Introduction

Increased efficiency is desirable in gas turbines from both an ecological and economic standpoint. Lower fuel consumption and lower emissions are key factors in the design of more efficient gas turbines. The leakage between the blades and the casing is one of the leading causes of losses in gas turbines [1]. Introducing a shroud, which is a platform attached to the blade tip, is one of the means of decreasing leakage. Usually, radial fins, called the labyrinth seal, are present at the shroud, further reducing the flow through the gap between the blades and the casing. The shroud and the labyrinth seal are integral parts of the blade, and they are manufactured together. In all modern engines, the geometry of the shroud is optimized to result in the lowest possible leakage, without compromising the structural integrity of the seal. This paper proposes a method for further decreasing leakage through fluidic sealing.

Turbine blades work under extreme conditions. They have to withstand significant heat (up to 1500 °C [2]) and high stresses from centrifugal and fluid forces. As a part of a rotating machine, they can be subject to vibration and fatigue damage [3]. Additionally, due to the combination of high temperature and stresses, creep can occur [4]. Therefore, the selection of materials and manufacturing technique is crucial for the durability of the blades and, in turn, the whole engine. Most of the blades in modern turbines are manufactured using superalloys via an investment casting process, which has limitations with respect to the minimum size of holes that can be introduced into the blade.

Leading turbine producers are already using 3D-printed blades in the newest operational turbine prototypes [4,5]. The use of 3D printing enables state-of-the-art materials to be used, such as titanium aluminide alloys, which have desirable characteristics for use under turbine operating conditions, with about 50% lower weight [6], but which cannot be used in manufacturing by traditional techniques.

The selection of material aircraft gas turbine engines is challenging even when considering most recent alloys. Turbine nozzles and blades are exposed to elevated temperatures and exhaust gases, causing surface degradation through corrosion and oxidation phenomena. Hot section components must ensure the required reliability and thermal stability throughout a long service life that exceeds one hundred thousand operating hours. Moreover, blades are constantly being subjected to severe mechanical loads.

To meet all of these requirements, materials require outstanding mechanical properties, high strength, long fatigue life and good resistance oxidation and corrosion at high temperature [7].

While titanium alloys are mostly used for the compressors of turbofan engines, the heat resistance of nickel-based superalloys makes them common materials for use in engine turbines [8].

Additionally, 3D printing makes it possible to manufacture complex shapes and hollow structures. This is especially desirable for high-pressure turbines, where blade cooling is necessary. Such cooling is carried out via a series of small holes supplied with cool air from the compressor. Moreover, the shroud and labyrinth seals of blades created using additive manufacturing can have more complex shapes. For example, the fins can be manufactured in such a way that additional jets are formed in the gap, creating fluidic sealing. Many fluidic sealing concepts have been patented and published over the years, starting with Smile and Paulson in 1960 [9,10,11,12]. However, they have never been implemented, and it is easy to assume that the difficulty of the traditional machining of complex seal geometries has played significant role in this decision process. Additive manufacturing makes it possible for old concepts to be revisited and optimized. A selection of concepts are summarized in the next paragraphs.

All of the configurations proposed over the years have introduced some form of jet or curtain in the labyrinth seal flow [13,14]. Many of the researchers proposed pressurized air from the compressor as a source for driving the jets [14,15]. The air from the compressor has to be supplied through some kind of system, and the use of compressed air should be taken into account when the effectiveness of such solutions is being assessed. Such an assessment has only been proposed by Auld et al. [14].

Since the pressure upstream of the blade cascade in a turbine is higher than downstream, the pressure difference from within the flow can be used as a source for the jets [16,17]. Such configurations do not have to be supplied from the compressor, and therefore, it may be simpler to introduce them into turbines. Additionally, such approaches use the pressure difference naturally occurring in the flow, and no additional energy has to be added to the system. Thus, compressing the additional air used in the seal does not have to be taken into account in the effectiveness calculations. A more extensive literature survey of the fluidic seals used in turbine shrouds can be found in [18].

The modification of the labyrinth seal presented in this paper is called the ‘air curtain’, and this modification is proposed in order to improve its effectiveness. The fluidic seal (curtain) is created in the gap between the fins of the seal and the casing by the high-velocity jet present in the bypass slot (Figure 1). The bypass channel connects the upstream face of the fin with the surface of the tip of the fin.

The effectiveness of the air curtain was on the basis of an experiment and numerical simulations. In the studied cases, the modification resulted in a 4–22% reduction in leakage, depending on the gap height and the pressure ratio [18,19].

The initial idea for the configuration presented was the result of 2D numerical simulations, but the resulting configuration was unfeasible, since part of the seal was not connected to the shroud. This paper presents a parametric study that was performed to modify the original 2D configuration in order to make it possible to use. In addition, the efficiency of the air curtain was studied as a function of the geometry of the bypass slot. While the configuration was studied from the fluid flow perspective, the constraints specifying the maximum and minimum dimensions took into account the structural and thermal aspects of the case. Additionally, the feasibility of manufacturing a blade with the air curtain slots was studied on the basis of the current state of the art.

## 2. Materials and Methods

### 2.1. Material Selection

There are several reasons for the wide application of nickel-based superalloys. Firstly, the content of nickel elements (weight percentage, wt.%) in the surface of the earth is higher than, for example, cobalt elements, while offering similar properties. A second reason is the crystal structure of nickel, which is a face centered cubic (FCC) structure, promoting moderate strength and ductility. Lastly, the significant mechanical properties related to the microstructures of Ni γ (matrix) and Ni_3_Al, γ’, the primary precipitation strengthening phase [7].

Nickel-based superalloys can be used in all stages of a turbine, from the hottest section near the combustor to the coldest at the exhaust frame.

In the most recent turbofan architectures, titanium-based alloys have been starting to take on a more and more important role in turbofan engines. The reason for this is weight reduction—the density of the titanium alloy (3.9 g/cm^3^) is half that of nickel-based alloys (8.3 g/cm^3^). At the same time, the maximum operating temperature of nickel-based alloys is 800 °C, compared to 1000 °C for titanium-based alloys; therefore, the former can only be used in colder parts of the engine.

The GEnx low-pressure turbine (LPT) introduces Titanium Aluminide blades to stages 6 and 7, reducing engine weight by approximately 150 kg (400 lbs) and contributing to increased fuel efficiency of the GEnx engine [19,20].

### 2.2. Casting and Machining

#### 2.2.1. Casting

The manufacturing process for nozzles and blades is long and difficult, and is explained in detail in [21].

Nickel-based superalloy remelting stock for investment casting is produced by vacuum induction melting (VIM) or other controlled-environment methods. Process atmosphere and material quality have a crucial impact on casting defectiveness, especially the level of porosity.

The casting process provides the possibility of controlling the grains. The most common structure is equiaxed grains. This process does not require any specific control. Grain structure depends on the solidification path, which usually starts at the thinnest blade profile, the trailing edge.

The newest turbine blades are made of single crystal [22]. They are widely used in high-pressure turbines, as well as in the hottest section of low-pressure turbines, where working temperature can increase up to as high as 1150 °C. No grain boundaries function as diffusion paths, therefore the creep resistance is improved. Another grain modification for producing blades is directional solidification. This structure consists of several columnar grains. The increase of properties is obtained by setting the boundaries mostly parallel to the major stress axis resulting from the blade’s working condition. 

The casting of TiAl alloys is like the process for standard equiaxed grains, but obtaining TiAl ingot is much more difficult due to the refining process of titanium ore and titanium’s affinity for oxygen. Moreover, TiAl reacts with all ceramics.

The requirements for sealing the fin onto the shroud have a negative impact on the castability of components by introducing locations where the walls have different thicknesses, which, in consequence, can lead to internal porosity and can affect gating system effectiveness.

The process in the foundry usually includes, in addition to investment casting, heat treatment and nondestructive inspections. The standard heat treatment operation is solutioning, which homogenizes the alloy structure. Castings are also controlled before machining operation by visual, fluorescent penetrant and radiographic inspection [21].

Another geometrical feature that would have a significant impact on the casting process is fluidic sealing slots, as shown in Figure 1. Depending on the size and geometry of the slots, there are several different methods for introducing them in the casting phase. The most practical one would be to use specially designed wax die inserts in the shape of the slot. Another approach involves adding additional elements, cores on wax or mold level. Casted air curtain slots would result in good surface finish characteristics without the need for additional machining operation. Nevertheless, the added challenges in the casting process could point to simple machining as being the most efficient method of producing the slots.

#### 2.2.2. Machining

Shroud machining is usually limited to grinding the surfaces that interlock (contact surfaces between two adjacent blades or labyrinth seals). The process is performed in 3-axis or 5-axis grinding machines using grinding wheels. Recent technologies make it possible to perform grinding operations with the correction automatically calculated by the machines. Dimensional process repeatability is extremely important from an engine assembly perspective. The tolerances for fins in the labyrinth seal are tight, and consider not only the tolerances of the blades, but also the tolerance of the casing and/or shroud (as separate components that are in contact with the blades).

Most airfoil components require the application of environmental coatings. Nowadays, there are plenty of coating deposition technologies on the market, enabling the formation of adhesive, diffusive and adhesive–diffusive coatings. Many of the coatings are evolutionally modified common technologies, for instance chemical vapor deposition (CVD), physical vapor deposition (PVD) or thermal spraying technologies [23]. The most common practice is to apply simple aluminide coating during the vapor process. The coating effectively protects components at up to 1080 °C in the oxidation, and corrosion type I and II temperature ranges. Considering different component architectures and machining routers, environmental coatings can impact the dimensions of fins by adding extra materials on the top surfaces.

Depending on the operating conditions and engine architecture, abradable coatings can be required in the casing [24]. The coating can be applied to opposite surfaces of separate shroud components. The tribo-system labyrinth coating requires the suitable selection of the rubbing partners [25], which is a wide topic that is outside the scope of this paper. 

### 2.3. Additive Manufacturing

#### 2.3.1. Printing

A final design is always a balance between optimized shape, costs and producibility. The proposed geometrical optimization of the labyrinth seal requires a new approach for manufacturing. The machining of bypass slots needs to be performed using a more developed process than standard grinding.

Fortunately, recent years have been groundbreaking in this field. Additive Manufacturing has seen continued growth on the market, redefining production and whole companies as a result of its impact on the supply chain, maintenance and component development. There are numerous Additive Manufacturing technologies offering various advantages and disadvantages. Methods that do not enable the creation of strong and thermally resistant parts were excluded, as those properties are vital. The strengths and weaknesses of the remaining technologies are summarized in Table 1. The selection of the most appropriate technology depends on a number of factors, where component geometry and material play an important role.

Considering recent trends in additive manufacturing, turbine blades were produced by Electron Beam Melting [27] as schematically presented in Figure 2 [28]. In the EBM machine, patented by the Swedish company Arcam, melting and consolidation of the layers is performed using the energy of an electron beam up to 3 kW. The entire process takes place in a high-vacuum chamber at a pressure below 10^−5^mbar. Electrons are emitted from a tungsten filament, accelerated through an anode to hit and melt the powder by the transformation of their kinetic energy into heat. The elevated build temperature is above 1000 °C for γ TiAl. Magnetic lenses are used to focus and control the movement of the beam. The build rate is up to 20 mm height per hour [27].

The advantages of Electron Beam Melting [27]:Significantly faster part manufacturing than for Laser Engineering Net Shape, SLS and SLMThe final part has low amount of impurities (H, O, N), due to process occurring in high vacuumProperties of part are better than the ones resulting from castingThe process is created so that the rapid solidification occurs, leading to fine microstructures of the partThe thermal stress can be released by rising the temperature of the process

However, the following risks associate with EBM also need to be considered:Rough surface finish compared with laser-based additive techniques. In consequence, stock materials on all surfaces need post machiningMicrostructure (duplex or lamellas structure, grain size and interlamellar spacing) and its impact on the ductility and the propensity for cleavage fractureSignificant plastic anisotropyLow recoverability due to low dislocation mobilityRelatively low diffusivitySlow recrystallization due to low grain boundary mobility

Given the availability of technologies that make it possible to obtain optimized blade fin geometries, an appropriate selection of material needs to be considered. Titanium, nickel and chrome–cobalt alloys can be used in the EBM process [26]. The benefits of titanium alloys over nickel alloys were already summarized in Section 2.1.

Some commonly used TiAl alloys and the influence of the alloying elements on each of those alloys are summarized in Figure 3 [28].

Based on its wide application in the aviation industry and the significant amount of experience in manufacturing with it using the EBM process [27], GE 48-2-2 seemed to be the most appropriate selection. This alloy consists of a Ti base alloyed with 48 at.% Al, 2 at.% Cr and 2 at.% Nb. The alloy offers increased toughness and ductility at lower temperatures, together with the desired creep properties compared to other γ TiAl [29]. The material for the EBM process must be in the form of a powder with a typical size of 45–105 μm. No binders or additives are necessary.

#### 2.3.2. Machining

After selection of the material and the manufacturing method, turbine blades still require additional machining operations to optimize their microstructure and dimensions. The left sample in Figure 4 shows the component after the additive process, but before machining. Before machining operation, the EBM parts are heat-treated.

Hot Isostatic Pressing (HIP) is a processing method in which both heat and high pressure are applied (so called “hipping”). Hipping is used to reduce/disperse the degree of porosity and homogeneous microstructure and remove the anisotropy within the material within the volume of the component [30]. Hipping is related to the solutioning process, because both are similar with respect to soaking phase time and temperature. Nevertheless, most manufacturing routes include both operations due to the cooling phase being crucial for β phase formation. In most cases, cooling is rapid. Standard hipping furnaces are not able to meet the requirements, and therefore a second heat treatment is applied in a vacuum furnace. 

Components manufactured by EBM require surface finishing. γ-TiAl alloys possess low thermal conductivity and brittle deformation; therefore, it is possible to introduce conventional machining processes, but it is not economical [31]. The state of the sample after conventional machining is presented on the right in Figure 4.

The alternatives to standard machining include unconventional processes: electro discharge machining (EDM) and electrochemical machining (ECM). ECM could be a capable manufacturing technology for γ-TiAl, as machining is independent of the mechanical properties of the processed material, in combination with high removal rates and no thermally or mechanically affected zones.

Precision Electrochemical Machining (PECM), which is a modified ECM used to manufacture spatial shapes, is an electrochemical cavity-sinking erosion process with oscillating electrodes and a regulated working gap. PECM applies a pulsed direct current between the electrode and the workpiece, which dissolves anodically in accordance with the geometry of the subsequent electrode [33].

The PECM process for finish-machining blade profiles uses two anode tools that simultaneously machine both sides of a profile using an oscillating motion combined with a pulsed power supply, as shown in Figure 5 [34].

The gap between the tool and the workpiece is much smaller than in ECM, decreasing from a few hundredths of millimeter to only a few micrometers, leading to a more effective arrangement of electrical field lines and improved machining accuracy. ECM/PECM offers 10- to 30-percent faster cycle times than conventional five-axis machining of blades, does not create burrs, and achieves fine surface finishes to 0.05 Ra, as shown in Figure 6 [34].

PECM can be adopted for the machining of most challenging geometrical features, including the slots for the air curtain. This process can be an alternative to standard drilling and electro discharge machining, possessing the following benefits:Lower cost per partNegligible electrode wearNo material hardness concernsWide material applicabilityUnlimited geometry.

As potential points for further development, the industrialization of PECM for slots should address the following issues:Reduction in steps such as shaping and deburring.Mismatches/steps in components dimensions are too big to proceed in one set up.Presence of intergranular attack or parent material cracking.

The detailed comparison of manufacturing technologies such as milling, EDM and ECM with respect to titanium- and nickel-based rough-machining operations was performed on the basis of a technological analysis focusing on the achievable material removal rates and an economic analysis for simplified gap geometry. ECM was chosen as the most cost-effective technology, especially in for large-scale production. Furthermore, via ECM, it is possible to achieve finished surface qualities during rough machining operations, which saves the need for further treatment like cost-intensive finish milling steps or polishing operations [35].

Based on the description of blade manufacturing presented above, labyrinth seal slot machining obviously has a strong relationship to the components. Nevertheless, considering only the slot requirement, the most effective approach for obtaining slots is additive manufacturing plus ECM:Additive manufacturing is recommended for TiAl alloys, which are preferable due to their low weight.Additive manufacturing makes it possible to produce slots with predefined geometries, which simplifies subsequent machining by limiting the amount of material that has to be removed.PECM makes it possible to obtain final shapes with unlimited geometry.PECM ensures the lowest impact on material.PECM is preferable from a technological and economic perspective.PECM does not required further surface finish.

While the producibility of the shroud with bypass slots was determined on the basis of a literature survey and professional experience, there is a need for further scientific study on the impact of slots on part durability and service life.

### 2.4. Numerical Model Definition

The investigated configuration is the seal of the last stage of the LP turbine of an aircraft engine. The geometry of the reference configuration is shown in Figure 7a. The distance between fins m is equal to 23 t, the fin height h = 12.5 t and the gap height s = 0.9 t. The shroud of the turbine blade has a width W equal to 70 t.

The sample geometry of the bypass slot can be seen in Figure 7b.

The values of the variables were constrained by the structural and thermodynamical aspects of the seal operation, and were determined in consultation with industrial partners from the INNOLOT/COOPERNIK project. Some of the studied cases fell outside the constraints proposed, thus permitting a wider view on the studied phenomenon. However, the final selection of parameters complied with the constraints.

For this study, the slot was placed in the first fin only. To accommodate the bypass slot, the thickness of the fin had to be increased from t to T = 3.25 t. The measurements show [36] that increasing the fin thickness (without introducing the slot) increases the leakage flow. Therefore, a seal with fins of thickness t were used as a reference case.

Four parameters were studied (the range is given in brackets): The slot outlet dimension in Z direction C (0.08–0.38 T)The slot inlet dimension in X direction B (0.2–1 T)The slot width A (0–1 L)The width of the slot segment L (0.05–0.33 W)

The dimension of the slot in the tangential direction was investigated in two of the studies. The first one (the slot width A study) tested which part of the fin should be covered by the slot. In other words, the L dimension was fixed at 0.33 W (shroud width), while A was varied between 0 L and 1 L. Therefore, this study assumed the use of three slot in one shroud. This is not guaranteed to be the best configuration. Therefore, in the next step, the L value decreased, with the A/L ratio kept constant. This study can be also thought of as a study to determine the most effective number of slots to use in one shroud.

The study presented here was a step in the preliminary phase of a proof-of-concept study on the air curtain modification. The full optimization had a computation cost that was too large for this stage. Therefore, the impact of one parameter on another was not studied. The parameters were changed in consecutive tests, as listed above. After each test, the value of the variables to be used in the following tests was chosen, based on the test results and the geometric constraints. During the initial tests, it was established that the outlet of the slot should be oriented in the direction countering the flow in the gap. Therefore, the outlet part of the slot is parallel to the downstream face of the fin, which is the maximum angle allowed.

Figure 8 shows an example of the computational domain with the boundary conditions marked. While the size of the domain varied between configurations due to differences in geometry, the boundary conditions were posed in the same way for all of the simulations. The boundary conditions were defined using atmospheric stagnation pressure and temperature. Additionally, a 5% turbulence intensity and a turbulent viscosity ratio of 5 were used at the inlet. The static pressure was set at the outlet to obtain a pressure ratio of 1.2. The validation case used a pressure ratio ranging from 1.05 to 1.85.

The RANS simulations were performed using Numeca FineTurbo software. The k-ω EARSM turbulence model was used [37], which proved the most accurate among the tested models. For the slot outlet dimension study (Section 3.2.1), the 2D model was used, while the remaining simulations used the 3D model. The perfect gas model was used, with viscosity defined using the Sutherland law. The explicit Runge-Kutta scheme and 2nd-order central difference discretization with artificial dissipation were set. Three levels of multigrid (Full Approximation Storage) and Merkle preconditioning were also used.

The simulations used structured, hexahedral grids. A grid convergence study was performed and is presented in [38]. The grid in the proximity of the side walls was created so that the y_+_ parameter was below 2, which allowed for good boundary layer prediction. The size of the grids differed between the cases, since the configurations of the models varied. For 2D grids (Section 3.2.1), the grid size varied between 90 and 120 thousand elements, while for 3D grids (remaining models) the size ranged between 0.7 and 2.7 million elements.

### 2.5. Numerical Model Validation

The numerical model is validated against the experimental measurements, which are described in detail in [38]. The leakage flow can be presented in form of discharge coefficient (C_D_). It is the ratio of measured mass flow divided by the isentropic mass flow:(1)$$C_{D}=\frac{{\dot{m}}_{meas}}{{\dot{m}}_{id}}$$
(2)$${\dot{m}}_{id}=\frac{p_{0}A}{\sqrt{T_{0}}}{\left\{\frac{2\kappa }{R\left(\kappa -1\right)}\left[{\left(\frac{1}{\pi }\right)}^{\frac{2}{\kappa }}-{\left(\frac{1}{\pi }\right)}^{\frac{\kappa +1}{\kappa }}\right]\right\}}^{\frac{1}{2}}$$
where *p*_0_ is stagnation pressure, *A* is the flow area, *T*_0_ is stagnation temperature, *κ* is ratio of specific heats for isentropic conditions, *π* is the pressure ratio and *R* is the gas constant.

Figure 9a shows the discharge coefficient for the reference case (without the air curtain). It can be observed that the leakage flow is underpredicted by about 10%; however, the trend is correctly predicted. The pressure distribution at the casing obtained in the simulations was also compared to the one from the experiment (Figure 9b), showing very good agreement. Overall, the accuracy of the numerical model is acceptable for simulating the flow through the labyrinth seal. The prediction of the C_D_ and pressure distribution depends on the turbulence model used. Several turbulence models were tested, and none of them produced accurate predictions for both C_D_ and pressure distribution. The impact of the turbulence model on C_D_ and flow structure was analyzed in [19].

## 3. Results

### 3.1. Parametric Study Results

#### 3.1.1. Slot Outlet Dimension in Axial Direction

Air curtain slots with two dimensions were tested (C/T = 0.08, 0.23 and 0.38). C/T = 0 represents the case with fin thickness increased to T, but no slot, while the dotted line shows the results for a fin with thickness equal to t. The same approach is used to present all of the cases in this paper.

Introducing even a small bypass slot decreases the leakage by about 1.5% (Figure 10). Increasing the slot dimensions initially leads to more leakage reduction (to 6%). Increasing the slot beyond C/T ~ 0.25 does not increase the leakage reduction. Beyond this point, the amount of the mass flow passing through the slot does not increase any more, which is a cause of no further leakage reduction. The constraints discussed in Section 2 allow a maximum slot outlet of 0.23 T, and this is used in further tests.

This case can serve as an example of the air curtain principle of operation (Figure 11). The pressure in the gap above the fin is lower than the pressure in the gap between the fin and the casing. Connecting those two regions with the bypass slot causes the flow through the slot. At the outlet of the slot, a jet is created that enters the area above the fin. This causes the blockage of the flow and the increase of the separation size in the gap. This leads to the reduction of the velocity of the flow in the gap and decreases the mass flow.

#### 3.1.2. Slot Inlet Height

Four dimensions of the slot inlet height were studied. Preliminary study found that the placement of the slot inlet on the fin does not have a significant impact on the efficiency of the air curtain, but the slot inlet should be placed as close to the fin tip as possible. This reduces the losses in the slot, making the jet slightly stronger. The minimum distance of the slot inlet from the fin tip allowed by the constraints is 1.2 T.

Five cases were compared. The first one was the slot without the inlet part. This case was prepared to study whether placing the slot in the fin without air supply impacts the mass flow. The results from this study showed that this impact was miniscule (below 1% mass flow difference). Further cases with the inlet height B varying from 0.2 T to 0.83 T. Results (Figure 12) showed an initial rise in air curtain effectiveness with increasing slot inlet size. Above the inlet height of 0.6 T, the leakage did not decrease significantly with further increase of slot inlet height. In this range, it is the outlet part of the slot that is the main limiter of the jet intensity, and the inlet part does not significantly limit the slot mass flow and impact how it affects the flow in the gap above the fin.

#### 3.1.3. Slot Width

In Section 3.2.1, 2D simulations were used to determine the slot outlet dimension. In this configuration, part of the seal was not connected to any support, which is impossible in real applications. Therefore, the slot is only able to cover part of the fin in the Y direction. Thus, the next step of the parametric study is to determine how much of the fin the slot should cover. The segment width L was fixed at 0.33 W.

In this study, the total width of the model was kept constant, while the slot width was increased from A/L = 0 (no slot) to A/L = 1 (full “2D” slot). The results are shown in Figure 13. Interestingly, the “2D” slot does not guarantee the maximum leakage reduction. The smallest leakage is obtained for a slot with A/L = 0.57; however, the cases with A/L = 0.36 and 0.79 also have similar results.

Slots that do not cover the whole fin are more effective in reducing leakage, because they introduce non-uniformity in the Y direction to the flow. Additional vortices are created at the edges of the slots (Figure 14), and the mixing in the cavern between the slots is amplified. This promotes kinetic energy dissipation, increasing the flow resistance and decreasing the flow. The issue was studied in more depth in [18].

Due to the geometrical constraints, the most efficient case (A = 0.57 L) was not able to be used, and a slot width of A = 0.36 L was chosen for further study.

#### 3.1.4. Segment Width

It was established that the slot should occupy around 0.36 of the fin width. However, this does determine what the absolute width of the slot should be. With the same A/L ratio, one can produce many slots with small widths, or one slot with a large width. Thus, simulations with a constant A/L ratio of 0.36 and a segment width varying between 0.05 W and 0.33 W were performed. This corresponds to installation of between 20 and 3 slots in a single shroud.

The largest leakage was obtained for the segment with the smallest width (Figure 15). This may be a result of the increased flow resistance in the slots with a small cross-section. A relatively large leakage was also present in the configuration with the widest segment. Here, it is likely that the mixing effects described in the previous section impact the flow less, since the number of streamwise vortices created is dependent on the number of slots, which is low in this case. The largest leakage reduction was obtained for L = 0.12 W, which roughly corresponds to eight slots in the shroud.

### 3.2. Width as a Variable for the Slot

#### 3.2.1. Introduction

The model constraints were established to make sure that the fins would not be physically damaged or overheat during operation. The part of the fin upstream of the slot has to be properly fixed to the rest of the fin; therefore, the slot should not take up more than half of the length of the fin in the Y direction. However, if the slot dimension A were not kept constant along its length, it would make it possible to use either inlets or outlets that were wider than 0.5 L. Such configurations were tested, and the results are shown in this section.

#### 3.2.2. Slot Inlet Width (Inlet Width Smaller than Outlet Width)

In this configuration (Figure 16), the outlet of the slot was kept constant (A2 = 0.8 L), which is a value larger than would be possible with a constant slot width. Meanwhile, the inlet width was varied between 0 (slot with not inlet part) and A1 = 0.8 L (constant slot width).

Figure 17 shows the dependence of the discharge coefficient on slot inlet width. The leakage decreases with decreasing inlet width. The most efficient configuration is the one with a constant slot width (largest slot inlet). This was unsurprising, since with a larger slot inlet, more air can be supplied to the jet. 

However, another comparison was made. Two cases with the same inlet width A1 = 0.4 L, but with different outlet widths, were compared. In this configuration, the amount of air supplied to the outlet part of the slot is similar. Even then, the case with a constant slot width is more efficient. Wider slot outlets create jets with a larger area, but which are significantly weaker with respect to both the area above the inlet and the area of low velocity (Figure 18).

#### 3.2.3. Slot Inlet Width (Inlet Width Larger than Outlet Width)

The configuration with an inlet width smaller than the outlet width exhibits a decrease in the jet intensity. The reverse configuration, where the slot inlet is wider than the outlet, was also tested, to see whether it intensifies the jet. The slot outlet width was constant A2 = 0.5 L, and the slot inlet varied from A1 = A2 = 0.5 L to A1 = L (Figure 19).

The cases where the slot inlet was wider than the slot outlet were more effective than the case with a constant slot width (Figure 20). The wider the inlet, the smaller the leakage. For the slot with the widest inlet, the leakage reduction was 150% compared to the reference case with constant width. Using a wide inlet fundamentally changes the flow field inside the slot. With a constant slot width, there is a separation at the edge inside the slot (Figure 21 and Figure 22). Due to the separation, the velocity of the jet in the middle, close to the upstream wall, of the outlet of the slot is smaller and the jet is less intense. The velocity at the sides of the slot is quite intense, due to the lesser degree of separation in the proximity of the wall caused by the lower velocity resulting from the boundary layer effect in the inlet part of the slot.

In the case with the wide slot inlet, there is no boundary layer effect at the inlet part of the slot, and the inflow to the outlet part of the slot is significantly altered. This separation occurs at three edges, and the jet core is stronger, although it is less intense at the sides. 

Interestingly, both cases have the same mass flow through the slot and the same average velocity in the slot. However, the jet with the stronger core, as a result of the use of a wide inlet, has a larger effect on leakage reduction.

While the configuration with the wide slot inlet is significantly better at reducing leakage, the use of a non-uniform width makes the slot more difficult to manufacture, and it is less structurally durable. Only a thin layer of metal holds the top and the bottom part of the fin. As this configuration was studied in the context of an experimental proof of concept, the more safe and simple, but less efficient, configuration was chosen. However, the interesting findings of this study will be very useful for preparing an optimal configuration when the solution is implemented.

Overall, an air curtain configuration that was feasible for manufacturing was established. As a result of tangential non-uniformity, it was more effective than the initial, unfeasible 2-dimensional configuration.

### 3.3. Mitigating the In-Slot Separation

#### 3.3.1. Introduction

The case in which the slot inlet was wider than the slot outlet (Section 3.3.3) revealed that the sharp edge inside of the slot causes the separation, leading to decreased jet intensity. However, the presented solution with a wide slot inlet is not feasible from a technological perspective. Therefore, other means of reducing the separation in the slot were investigated. These include the introduction of a chamfer or rounding on the edge, and changing the angle between the inlet and outlet parts of the slot.

#### 3.3.2. Rounded/Chamfered Slot Edge

An obvious way to decrease the slot separation is to make the transition from the inlet part of the slot to the outlet part of the slot smoother. This can be done using either rounding or a chamfer on the slot edge. Both of those options were tested with chamfer and a rounding size of c = 0.1 T (Figure 23).

Both configurations show a decrease in the leakage through the slot of about 2% compared to slots with a sharp edge. It can be observed in Figure 24 that the separation inside the slot is limited, and the jet is more intense.

#### 3.3.3. Inclined Slot Inlet

Another way to limit the separation is to increase the angle between the horizontal (inlet) and vertical (outlet) part of the slot. The base angle in the slot is 90°, and this was changed to 115 and 140°. With an increase in the angle, the leakage through the gap decreases by 2.5 and 3.5%, respectively, compared to the 90° case (Figure 25). This means that for a slot angle of 140°, the reduction with respect to the reference case without the AC is 50% greater than in the 90° case.

The separation inside the slot is weaker with the larger angle. Comparing the 90° case with the 140° case, one can observe that the mass flow through the slot is 9% higher for the latter. Figure 26 shows that the jet is wider in the cases with increased angles. With an increased angle, the separation at the slot inlet increases. It can be reduced via further changes in the shape of the slot.

Overall, it can be seen that the shape of the slot can be improved to increase the efficiency of the air curtain. Sharp edges should be avoided, and the transition from the inlet to the outlet part of the slot should be smooth, in order to prevent separation inside the slot. This paper presents part of the proof-of-concept study for the air curtain, and therefore, in this case, simplicity is more important than maximum efficiency. Further study on the air curtain will be aimed at creating the optimal shape of the slot.

### 3.4. Air Curtain Used in Both Fins

The first step of the study used the air curtain slot in the first fin only. Once the final geometry was chosen, the AC was also implemented in the second fin. The relative placement of slots in the first and the second fin was studied in order to investigate whether it has an impact on the leakage flow.

Two configurations were tested (Figure 27). In one, the slots on the first and the second fin were placed in line (aligned). In the second configuration, the slots on the second fin were placed so that the slot was in the same “tangential” direction as the gap between the slots on the first fin (staggered).

For this study the thickness of both fins have increased to T = 3.25 t. This causes the leakage increase by 13% with respect to reference case with fins with thickness t.

There are two ways of selecting reference cases for air curtain comparison. The first way, which has been used thus far, uses a baseline case with a fin thickness t as a reference. In this way, the concept is compared to the configuration that is used in the turbine. The other method is to compare the case with AC to the case without AC using the same blades but with a thickness of T. This comparison shows the potential of the air curtain, assuming that the unmodified fin geometry allows for the installation of the slots.

Installing the AC in the first fin results in reduction of leakage by 5%, when compared to the seal with fins with a thickness of t. Introducing the AC on the second fin doubles the leakage reduction (11%). On the other hand, when using the case with a fin thickness of T as a reference, installing the AC on the first fin reduces the leakage by 16%, and installing it on both reduces the leakage by 21% (Figure 28). This means that introducing the AC on the second fin does not improve the efficiency as much as adding it on the first fin.

There is a slight difference (below 0.5%) in mass flow between the aligned and staggered cases. In the aligned case, the high-velocity fluid exiting the area between the slots in the first gap directly enters the area between the slots of the second gap. In the staggered case, the high-velocity fluid exiting the first gap deviates, to enter the zone between the slots in the second fin. However, the deviation is slight due to the large distance between the fins, and therefore it does not generate significant additional losses. If the air curtain were to be installed in the turbine, the impact of rotation velocity on the interaction between the high-velocity zones created in each gap should be studied, to investigate whether it has an impact on the seal efficiency.

## 4. Summary

This paper presents an analysis of the manufacturing technologies used for producing the turbine blades, both via casting and additive manufacturing. The feasibility of manufacturing air curtain slots using both technologies was discussed. Additive manufacturing allows for more freedom in the selection of slot geometry, and introducing slots using this technique does not make the manufacturing of the blade significantly more complex. On the other hand, introducing the slots in the cast blade is a more complex process, requiring additional wax die. However, the internal surface of the slot does not require machining, which may not be the case for blades created using additive manufacturing. In slots created during casting, geometrical limitations resulting from the nature of the technology are to be expected. The technological aspects of the introduction of the air curtain in the labyrinth seal were studied based on a literature survey and our professional experience; further experimental or numerical study is necessary for conclusive assessment of durability and service life.

The air curtain is a flow control technique that decreases the leakage though the turbine shroud sealing. This paper presents a parametric study on the dimensions of the bypass slot, which is a crucial part of the air curtain. The inlet and outlet dimensions of the slot were studied, as well as the most effective slot width. The research showed that the outlet dimension in the Z direction C (along the flow) should be about 0.23 of the fin thickness T. The slot inlet height B should be above 0.6 T; however, increasing it beyond this value does not result in a radical increase in effectiveness. It was established that the slot is not at its most efficient when it covers the whole span (in the Y direction) of the fin, but is limited to about 0.4–0.8 L. This is fortunate, since due to limitations in the manufacturing process, it cannot cover the whole span. The increased effectiveness for the slot covering only part of the span is caused by the creation of additional vortices by the bypass slot. The vortices intensify mixing, which limits leakage flow. The last parameter studied was segment width L, which is an inverse of the number of slots in one shroud of a turbine. It was established that one segment should cover about 0.12 of the shroud width, corresponding to about eight slots in a shroud. This is a middle ground between creating more of the aforementioned vortex structures (more slots leads to more vortices) and the flow resistance in the slot (which is larger for smaller slots).

The final configuration shows a leakage reduction of 7% with respect to the reference fins with thickness t, and 17% with respect to fins with thickness T. This configuration corresponds to the case with an air curtain used on the first fin only, for a single (relatively small) gap height and pressure ratio. Other studies indicate that the effectiveness of the air curtain increases with increasing gap height.

Additional parameters, including the variation of the slot width and modifications of the slot angles and edges, were also studied. Additionally, air curtain slots were introduced in the second fin, and the relative positions of the slots were tested. The results of those studies will be presented in the extended version of this paper.

## Figures and Tables

**Figure 1 materials-14-03477-f001:**
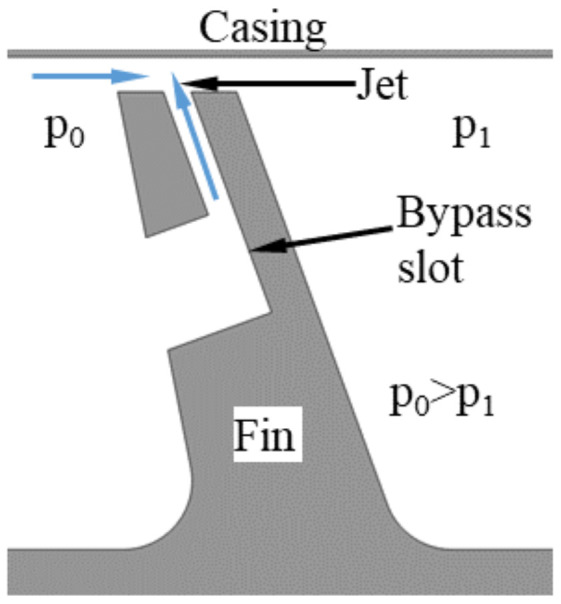
Air curtain schematics.

**Figure 2 materials-14-03477-f002:**
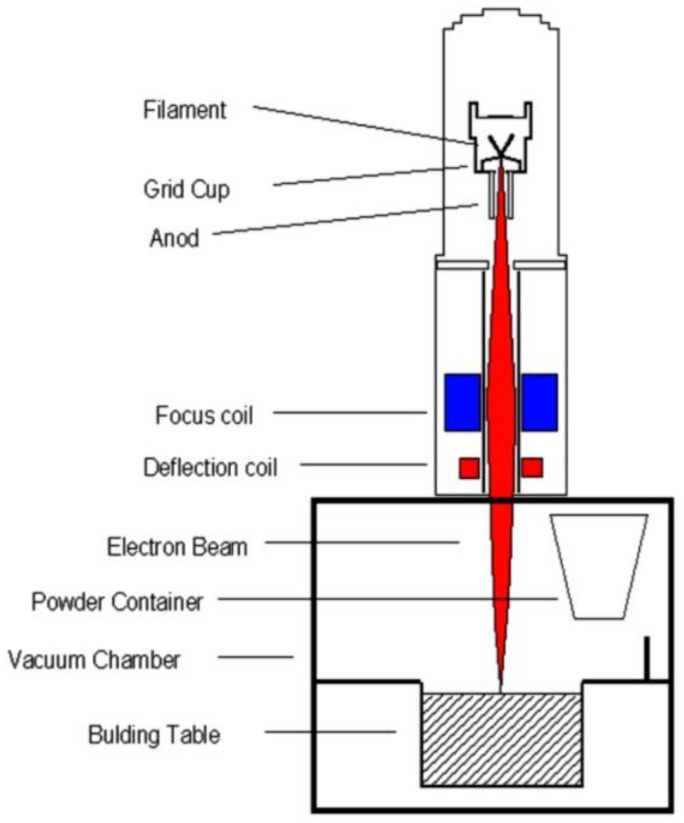
Layout of the electron beam gun column [28].

**Figure 3 materials-14-03477-f003:**
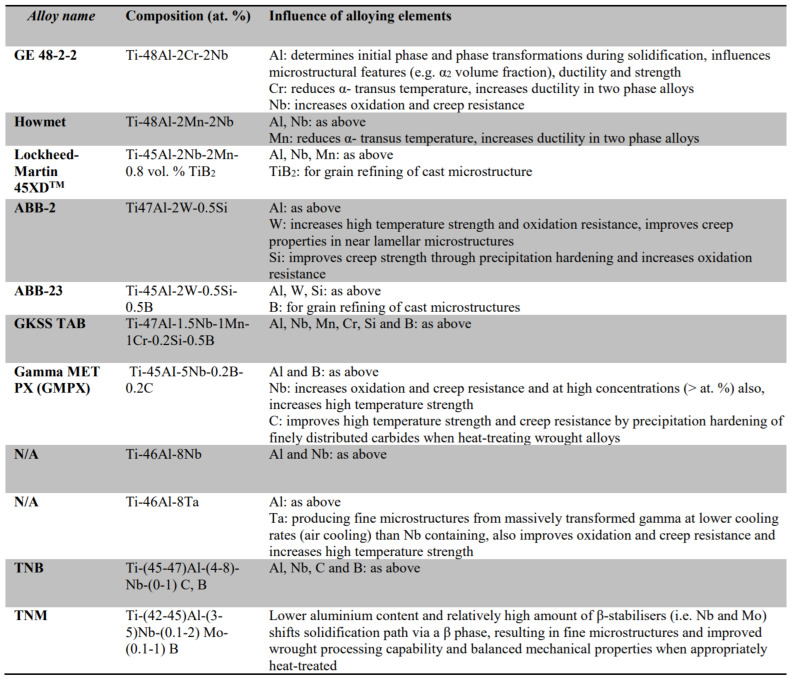
Summary of commonly used TiAl alloys and influence of alloying elements [28].

**Figure 4 materials-14-03477-f004:**
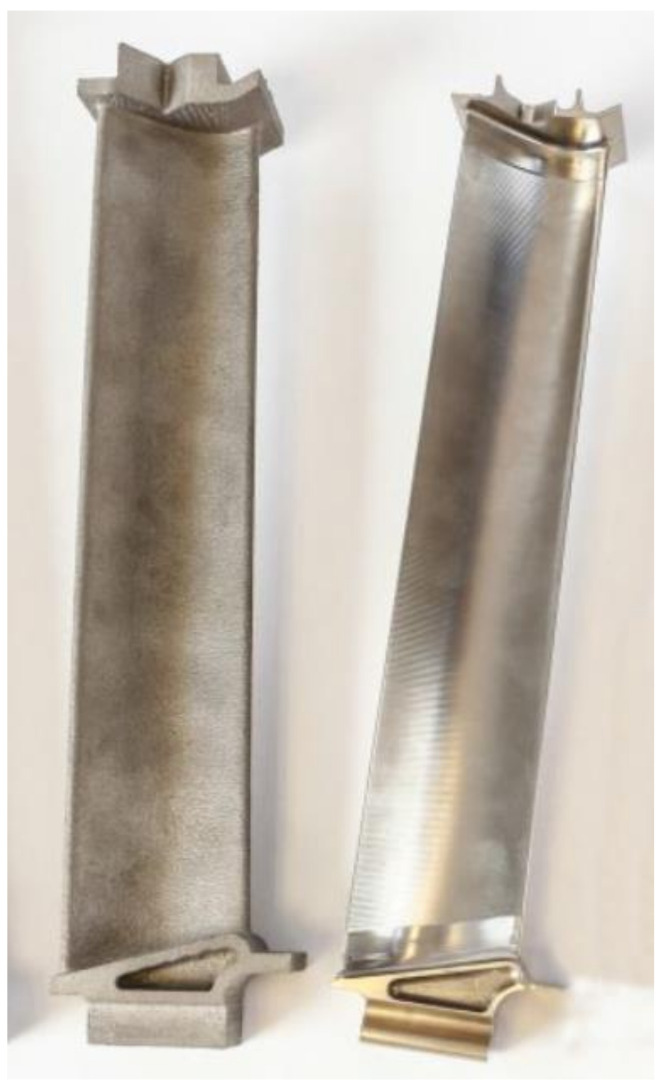
Comparison of turbine blades manufactured by EBM before and after conventional machining © 2018, Avio Aero [32].

**Figure 5 materials-14-03477-f005:**
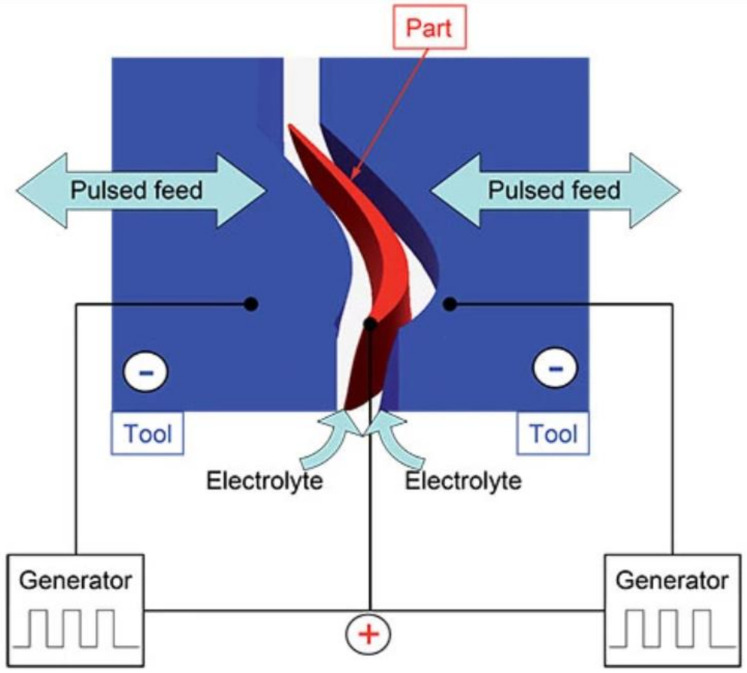
Schematic PECM process for turbine blades [34] © EMAG GmbH & Co. KG, 2021.

**Figure 6 materials-14-03477-f006:**
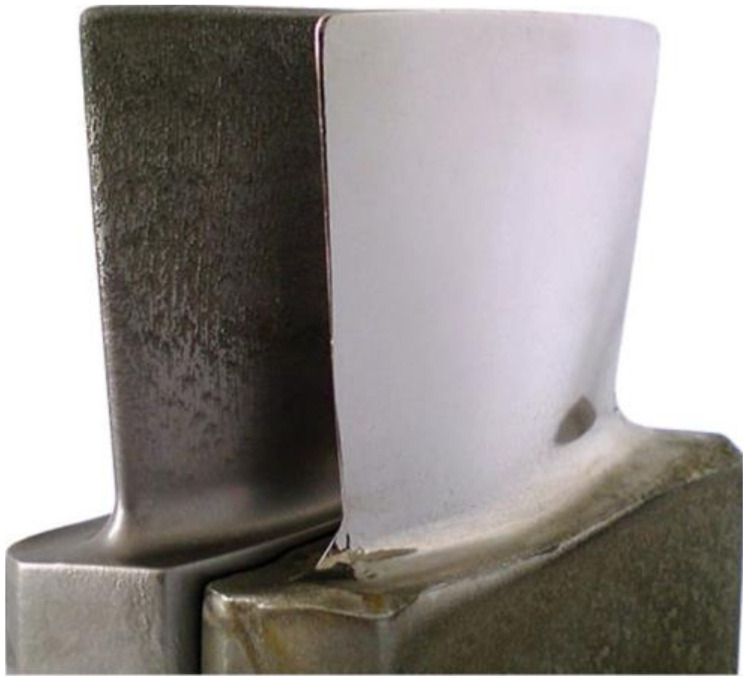
Surface finish after rough machining (left) and finish machining (right) [34] © EMAG GmbH & Co. KG, 2021.

**Figure 7 materials-14-03477-f007:**
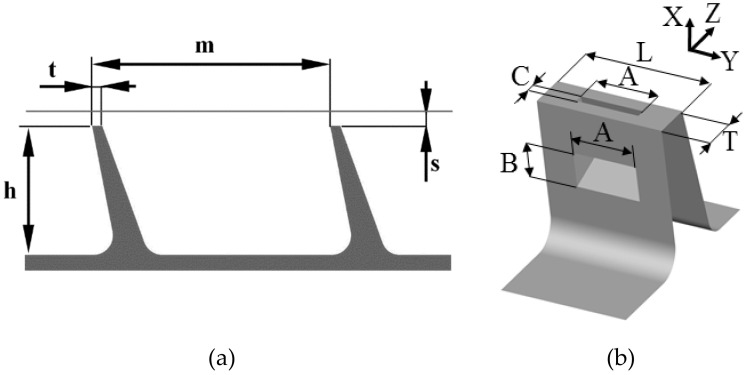
(**a**) The geometry of the reference labyrinth seal; (**b**) air curtain slot dimensions.

**Figure 8 materials-14-03477-f008:**
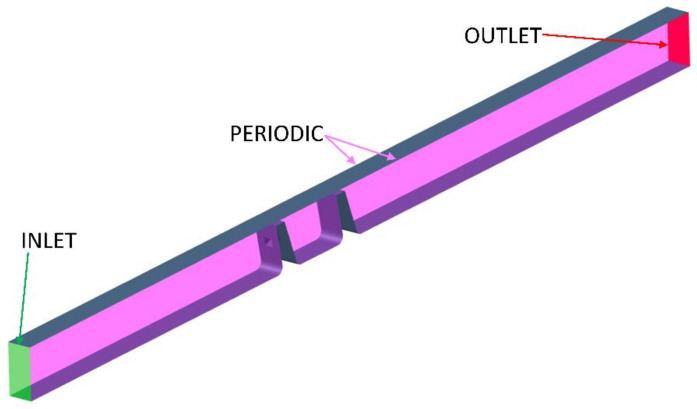
Sketch of the computational domain.

**Figure 9 materials-14-03477-f009:**
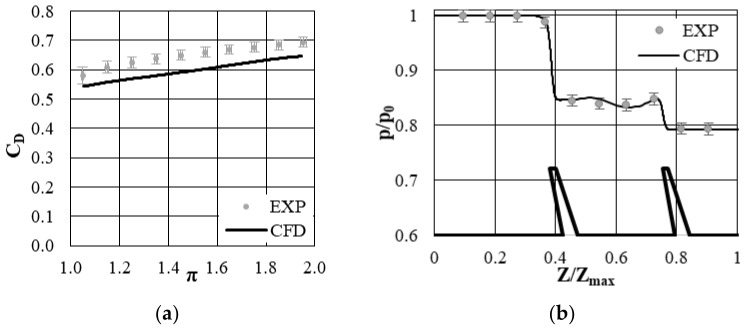
(**a**) Discharge coefficient versus pressure ratio, experiment and simulations. (**b**) Pressure distribution at the casing, experiment and simulations.

**Figure 10 materials-14-03477-f010:**
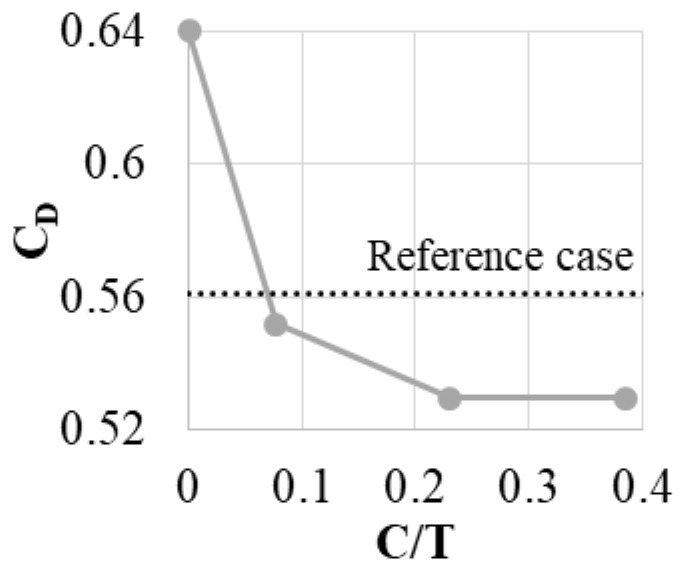
Discharge coefficient varying with changing C dimension.

**Figure 11 materials-14-03477-f011:**
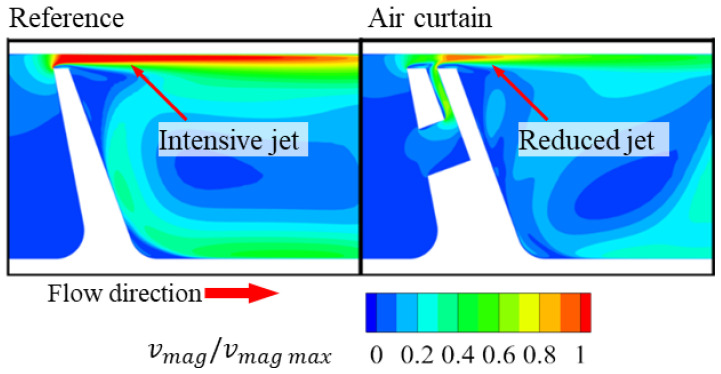
Example of the air curtain principle of operation, based on normalized velocity contours.

**Figure 12 materials-14-03477-f012:**
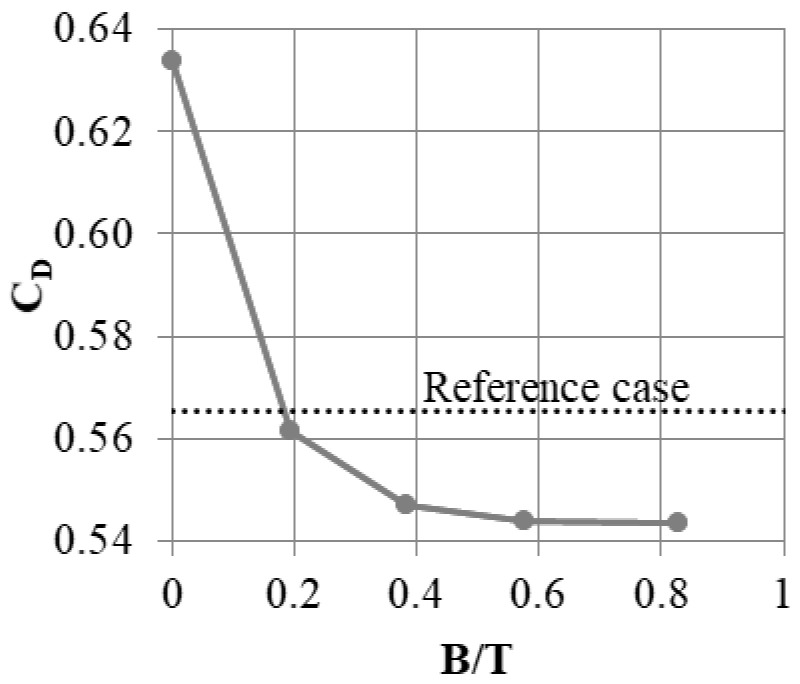
Discharge coefficient change with varying slot inlet height B.

**Figure 13 materials-14-03477-f013:**
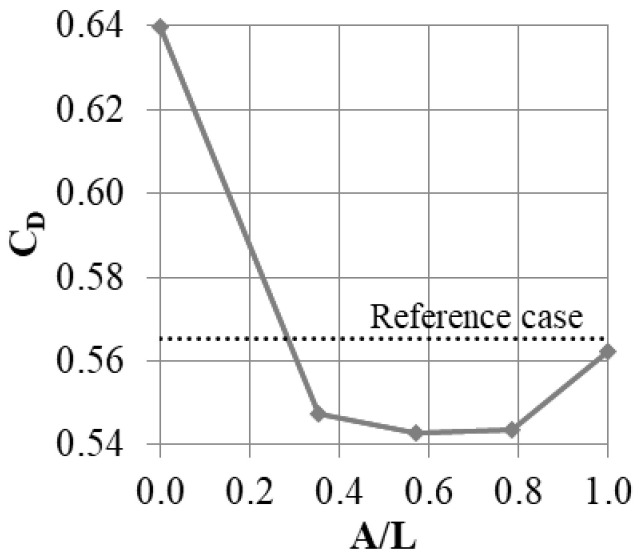
Discharge coefficient change with varying slot tangential direction.

**Figure 14 materials-14-03477-f014:**
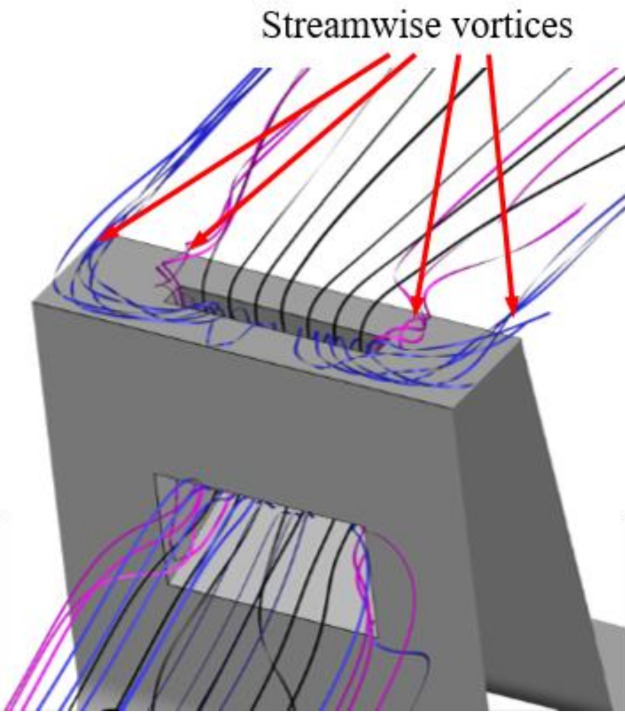
Example of vortical structures created be the presence of the air curtain slot.

**Figure 15 materials-14-03477-f015:**
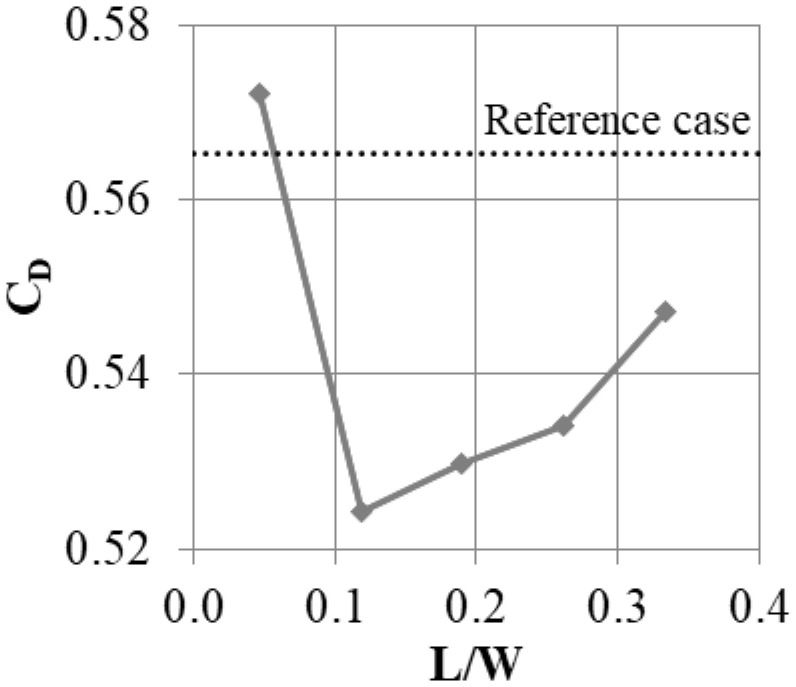
Effect of the segment width L on the leakage mass flow.

**Figure 16 materials-14-03477-f016:**
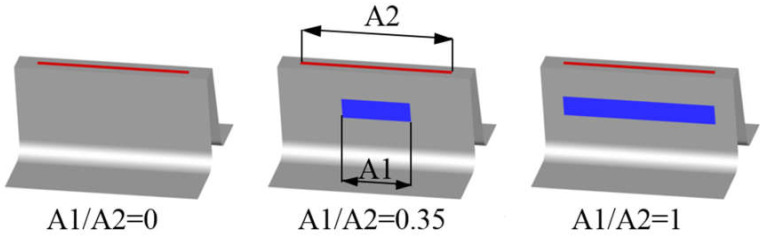
Example of variable slot inlet width.

**Figure 17 materials-14-03477-f017:**
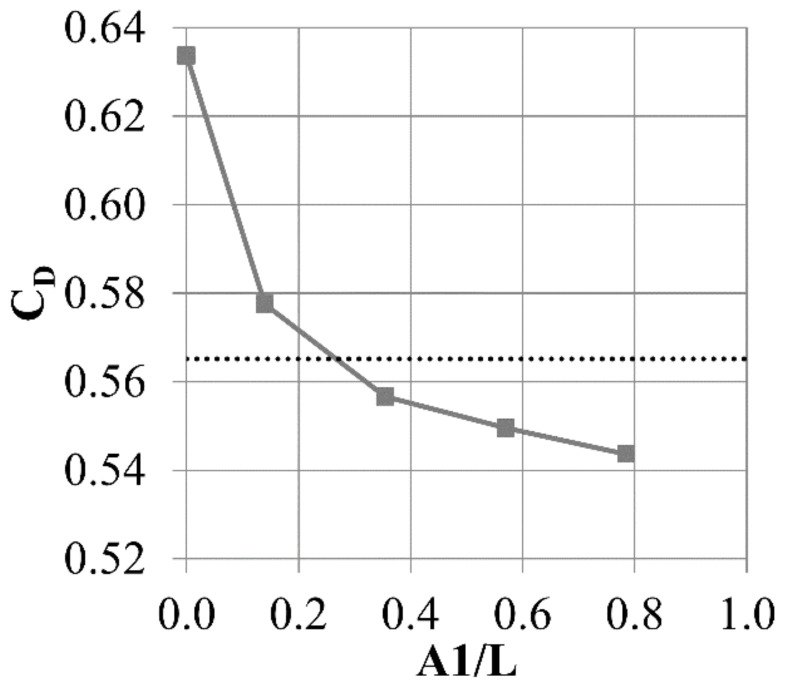
Discharge coefficient change with varying slot inlet width.

**Figure 18 materials-14-03477-f018:**
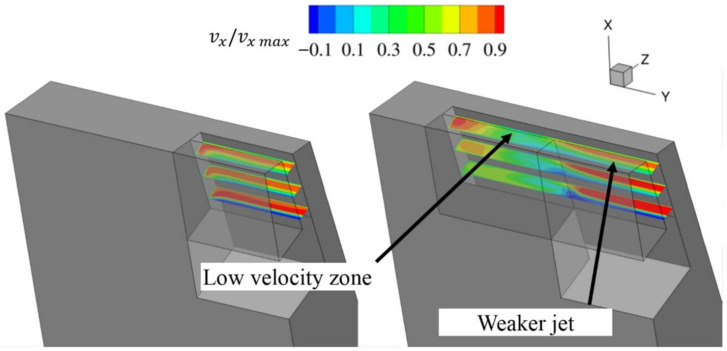
X velocity in the slot.

**Figure 19 materials-14-03477-f019:**
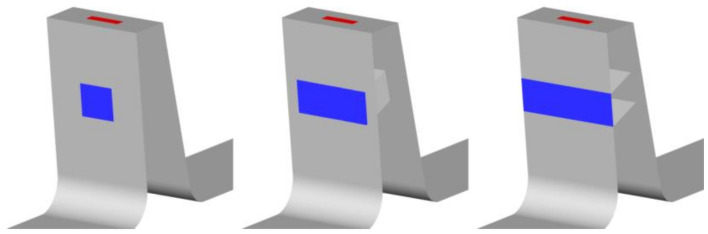
Example of variable slot inlet width.

**Figure 20 materials-14-03477-f020:**
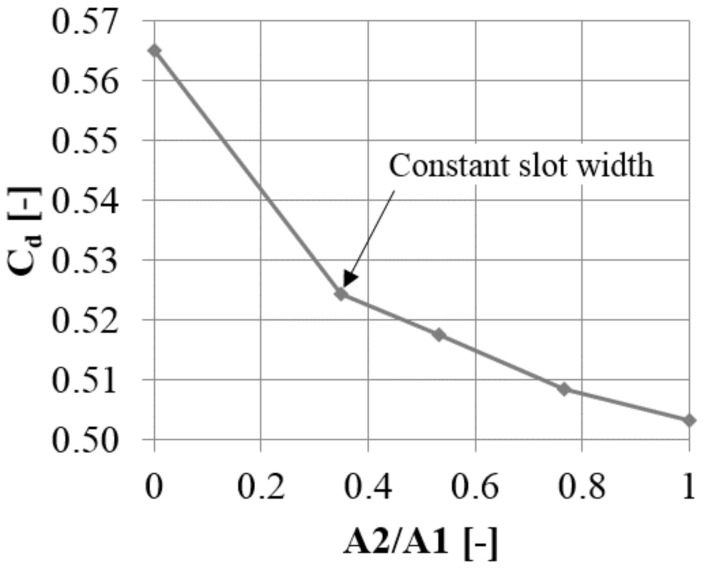
Discharge coefficient change with varying slot inlet width.

**Figure 21 materials-14-03477-f021:**
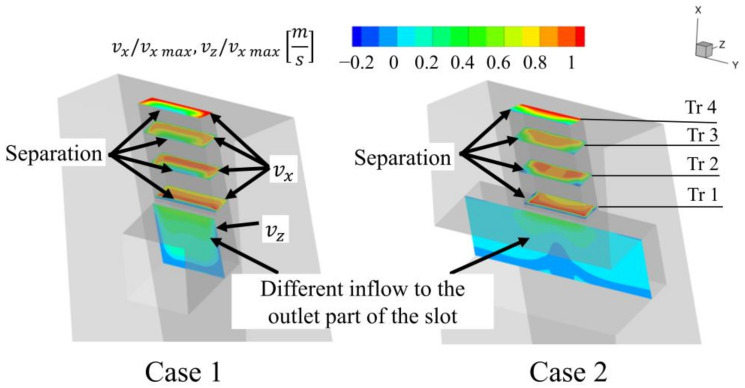
Flow structure in the slot for varying inlet width (Z velocity in the horizontal part of the slot, X velocity in vertical part of the slot.

**Figure 22 materials-14-03477-f022:**
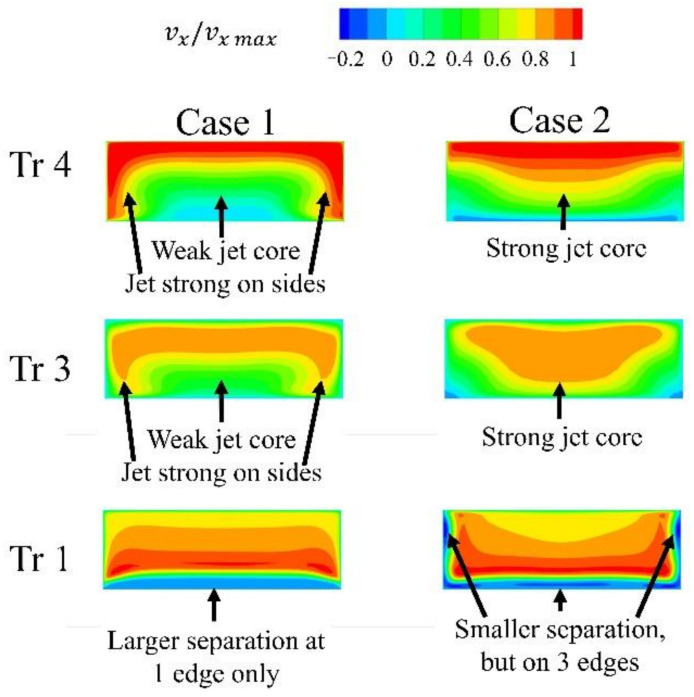
X velocity in the slot.

**Figure 23 materials-14-03477-f023:**
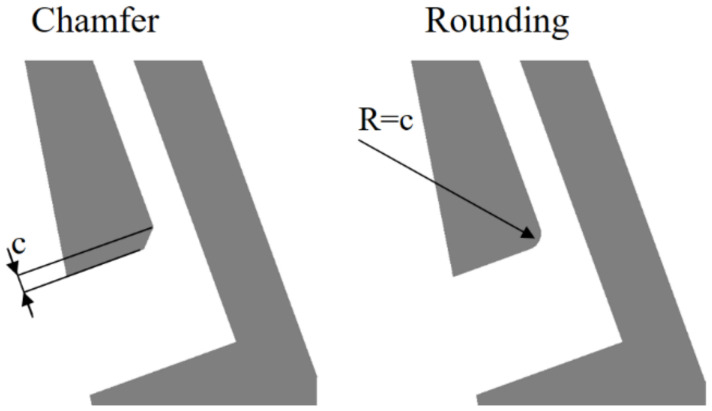
Example of rounding and chamfer.

**Figure 24 materials-14-03477-f024:**
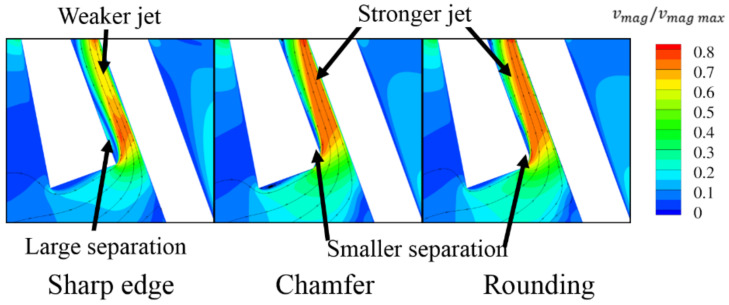
Velocity magnitude and streamlines for sharp, rounded and chamfered edge.

**Figure 25 materials-14-03477-f025:**
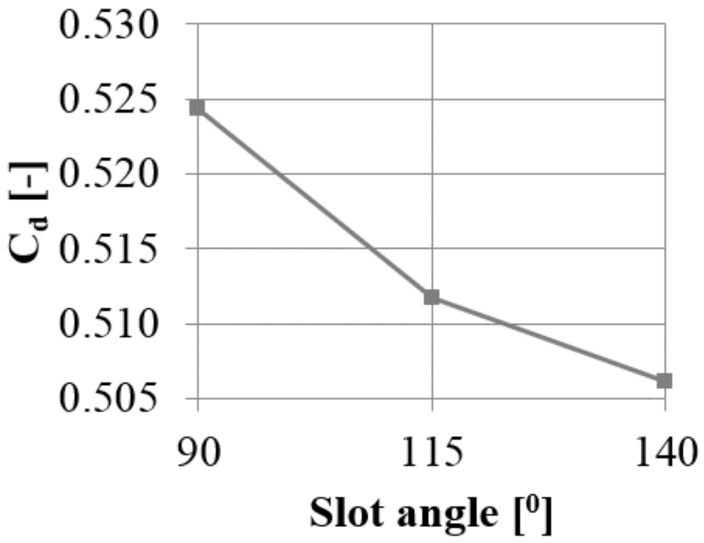
Discharge coefficient change with varying slot angle.

**Figure 26 materials-14-03477-f026:**
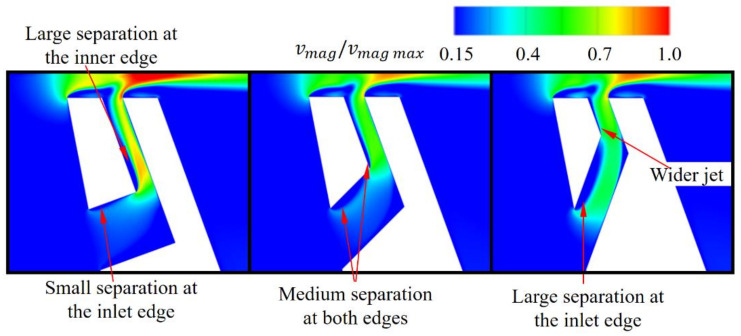
Flow structure in the slot with varying angle.

**Figure 27 materials-14-03477-f027:**
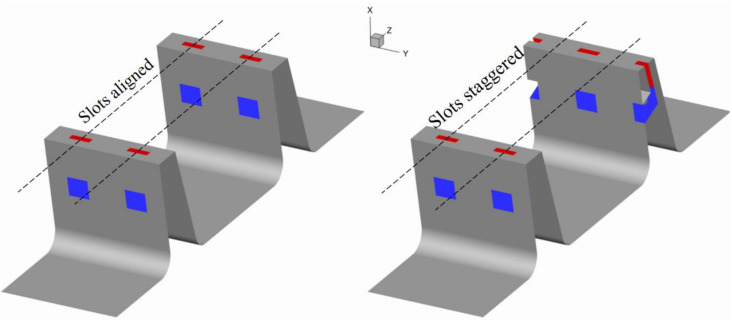
Location of slots on two fins.

**Figure 28 materials-14-03477-f028:**
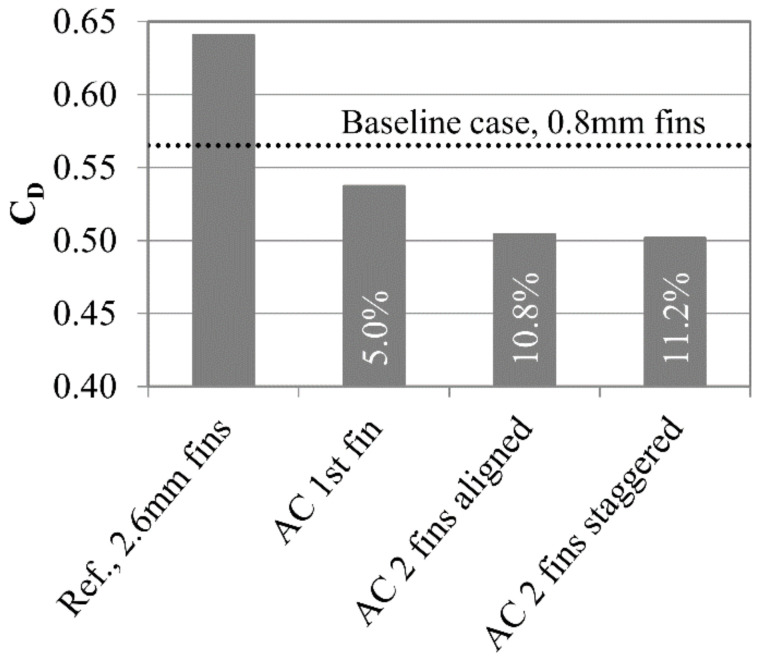
Leakage for various 2 fin configurations (leakage reduction in the bars).

**Table 1 materials-14-03477-t001:** Characteristics of selected additive manufacturing technologies (based on [26]).

Method Feature	Material Jetting	Electron Beam Melting	Laser Engineering Net Shape	Selective Laser Sintering	Selective Laser Melting
Manufacturing Time	+	-	+	-	-
Surface Quality	+	+	-	-	-
Mechanical properties	+	+	+	+	+
Thermal Properties	+	+	+	+	+
Complexity of parts	+	+	+	+	+
Manufacturing Dimensions	+	-	+	+	-
Precision	+	+	-	+	+
Post Processing	-	-	-	-	-
Operation Cost	-	-	+	-	+

## Data Availability

The data presented in this study are available on request from the corresponding author. The data are not publicly available since parts of it are IP of Avio Aero.

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
