# Peer review of "Feasibility Study of Fluidic Sealing in Turbine Shroud"

_materials, 2021, doi:10.3390/ma14133477_

Round 1

Reviewer 1 Report

See the attachment.

Author Response

The authors would like to express gratitude to the Reviewer for his/her comments. We think that they make the paper better. The text has been checked again and language and editorial mistakes were corrected. The responses to the specific questions were uploaded in a pdf file.

Reviewer 2 Report

The paper is divided into 2 parts, one dedicated to the manufacturing processes of turbine blades and another one to flow simulations when different bypass slot geometries.

The problem is that those two parts are not related and that although authors claim to have discussed "the feasibility of manufacturing the ... slots", the text does not contain any specific investigation regarding the possible problems associated with the manufacturing of these slots. They provide only a general description of the manufacturing processes of blades.

The authors should consider adding a description of the advantajes and drawbacks of having slots in the blades when using different manufacturing techniques or to, simply, delete that part and prepare the paper for a more suitable journal as no new information about materials is presented in the paper.

Regarding english, the paper needs another check, as some sentences must be amended. E.g.:

  • Line 147: "...operation is solutioning which homogenize..."
  • Line 416: "...what should be the absolute width of the slot should be"

Author Response

The authors would like to express gratitude to the Reviewer for his/her comments. We think that they make the paper better. The text has been checked again and language mistakes were corrected.

As to the main point of the review, the authors understand the Reviewer’s point of view. The main original part of the paper is centered around the fluid flow aspect of the labyrinth seal. However, one of the topics listed in the scope of the special issue the paper was submitted to is:

“Eco-innovation systems for mechanical engineering of equipment machines, renewable energy installations, recycling, and material processing;”

Our idea was inspired by the scope of the special issue, since presented research improves the efficiency (which reduces fuel consumption and pollution) of widely used gas turbines, but at the same time materials aspects of the design are really important.

We have included a wide introduction, focusing on the production of labyrinth seals, to present the topic from the perspective more familiar to the Materials journal readership. Even though no original research connected to material sciences is presented in the paper, we think that the novel concept of fluidic sealing in the form presented in the paper raises important research questions from the material science perspective. In fact the concept would be non-feasible, were it not for the progress in the field of precise machining and additive manufacturing. We hope that it can spark a discussion or research focusing on producing small scale elements/orifices in the labyrinth seal, so that more flow control methods can be considered to limit the leakage flow in a turbine.

Round 2

Reviewer 2 Report

As the special issue “Eco-innovation systems for mechanical engineering of equipment machines, renewable energy installations, recycling, and material processing;” as one of its topics, the paper can be accepted for publication. Other corcerns have been corrected.